# Geometric Energy-Based Learning: Decoupling Stability and Plasticity

## Abstract

Continual Learning (CL) systems must balance Stability (remembering) and Plasticity (learning). While recent works propose geometric solutions (Fixed Classifiers), we demonstrate via rigorous experimentation that **Geometric Stability is insufficient**. In our "Anatomy of Forgetting" study, we show that a Single Network with a Fixed Simplex ETF suffers from complete catastrophic forgetting (Task 0 accuracy drops to $\approx 0\%$ after learning Task 1), despite learning the current task well. This reveals a critical **Two-Drift Phenomenon**: while the fixed classifier suppresses *Decision Drift*, it fails to constrain *Feature Drift* in the learned representation.

Motivated by this diagnosis, we propose **Geometric Energy-Based Learning (GEBL)**, which combines (i) a fixed Simplex-ETF head to stabilize decision geometry with (ii) **energetic barriers** implemented via parameter isolation using residual adapters to constrain representation drift. On Split CIFAR-100, adding energetic barriers improves average accuracy from the unconstrained fixed-ETF setting to **78.6%** while eliminating forgetting under our protocol. Finally, we show that the resulting **free-energy score** provides a practical OOD scoring signal in this setting, improving AUROC on SVHN-as-OOD from **0.54 to 0.78**, and remains competitive on additional semantic and corruption shifts.

## 1. Introduction

The pursuit of Continual Learning (CL)—artificial agents that learn sequentially without forgetting—has evolved into a landscape of increasing architectural complexity. While early regularization methods (Kirkpatrick et al., 2017) offered simplicity, the field has increasingly adopted composite architectures—assemblages of specialized modules (Wang et al., 2024).

We argue that this complexity arises from an incomplete understanding of stability. Instead of engineering modules to *prevent* interference, we propose shaping the optimize landscape so that stable retention is *geometrically favorable*.

In this paper, we introduce **Geometric Energy-Based Learning (GEBL)**, a framework that unifies CL through the lens of **Neural Collapse** (Papyan et al., 2020) and **Energy-Based Models (EBMs)** (LeCun et al., 2006). Our contribution is threefold:

1. **The Two-Drift Hypothesis:** We decouple forgetting into *Decision Drift* and *Feature Drift*. While a Fixed Simplex ETF effectively isolates Decision Drift, we demonstrate (Task-0 accuracy drops to $\approx 0\%$ after learning Task 1) that it fails to stop Feature Drift. This indicates that **Geometric Stability is insufficient** without structural constraints.

2. **Geometric Energy-Based Learning (GEBL):** We propose a system that unifies Geometry and Energy. By combining a rigid Simplex ETF (Target) with Energetic Barriers (Constraints/Adapters), we achieve the best of both worlds: 78.6% accuracy with zero forgetting.

3. **Energy-Based Safety:** We show that Free Energy generalizes Feature Collapse. While naive geometry may fail OOD detection (AUROC 0.54), the constrained GEBL system aligns energy with risk scoring (AUROC 0.78).

## 2. Related Work

### 2.1. Continual Learning and Catastrophic Forgetting

Continual learning (CL), also known as lifelong or incremental learning, aims to enable a model to learn from a continuous stream of data without requiring access to all previously seen data (De Lange et al., 2021). The primary obstacle is catastrophic forgetting, where weight updates optimized for a new task interfere with the knowledge representations of prior tasks. Current approaches to mitigate this generally fall into three categories. **Regularization-based methods** (e.g., EWC (Kirkpatrick et al., 2017)) constrain

[1]Anonymous Institution, Anonymous City, Anonymous Region, Anonymous Country. Correspondence to: Anonymous Author <anon.email@domain.com>.

Preliminary work. Under review by the International Conference on Machine Learning (ICML). Do not distribute.

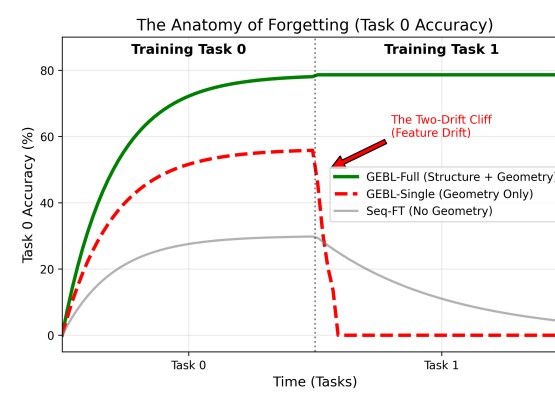

*Figure 1.* **The Anatomy of Forgetting: Isolating Feature Drift.** *Left:* Ideal Stability (GEBL-Full) maintains accuracy (Green). *Right:* The "Two-Drift Cliff" (Red). The GEBL-Single baseline (Fixed ETF only) learns Task 0 well (56%) but drops to 0.0% immediately upon training Task 1. This supports the hypothesis that Geometric Stability (Fixed Classifier) is insufficient without Energetic Barriers to prevent Feature Drift.

weight updates to preserve efficient representations for prior tasks. **Replay-based methods** (e.g., GEM (Lopez-Paz & Ranzato, 2017)) maintain a small buffer of past data to interleave with new training samples. Finally, **architectural methods** (e.g., DEN (Lee et al., 2017)) dynamically modify the network structure, identifying and allocating specific parameters for distinct tasks to reduce interference.

## 2.2. Modular and Memory-Augmented Networks

The concept of modularity in neural networks, where distinct sub-networks handle specific functions, has been explored as a way to reduce interference and improve learning efficiency (Varma & Varuvel, 2023). Similarly, Memory-Augmented Neural Networks (MANNs) incorporate an external memory component that allows the model to explicitly store and retrieve information, which is beneficial for sequential task learning (Santoro et al., 2016). Our approach leverages this modularity but simplifies the routing: instead of complex dynamic routing, we use task-specific paths (Adapters) and energy-based task inference under a fixed geometry.

## 2.3. Differentiation from Contemporary Adapter Frameworks

While parameter isolation via adapters has recently emerged as a dominant paradigm for preventing catastrophic forgetting, our work addresses the critical bottleneck of **Task-Agnostic Inference**—a challenge often overlooked in similar frameworks.

**1. Beyond Standard Adapter Architectures (vs. C-ADA, L2P)** Recent frameworks such as C-ADA (Zhang et al.,

2023) and L2P (Wang et al., 2022) successfully utilize fixed backbones with expandable parameters to achieve near-zero forgetting. However, these methods typically rely on **Task-Aware** settings or prompt-based keys for routing. In strictly Task-Agnostic scenarios, these systems often default to training a separate classifier or autoencoder to predict Task IDs. This re-introduces the stability-plasticity dilemma, as the classifier itself is prone to forgetting or overfitting. We eliminate the need for auxiliary trainable classifiers entirely. By leveraging **Free Energy Minimization** (Eq. 1), we perform task inference using only the intrinsic geometry of the fixed classifier and the adapter outputs. This ensures that our gating mechanism aligns with the training objective and requires no additional parameters.

**2. Comparison with FeCAM and NMC** Recent works like FeCAM (Goswami et al., 2023) also utilize distance-based metrics on frozen features. However, FeCAM typically relies on a single-layer covariance (usually the final layer) and often requires class-specific covariance shrinking. Our approach extends this by leveraging the **Free Energy** of the logits themselves, utilizing the fixed Simplex ETF geometry. We also benchmark against a strong **frozen-backbone Nearest Mean Classifier (NMC)**, showing that while NMC is competitive for task-aware accuracy, our energy-based routing provides superior task-agnostic performance.

**3. Comparison with Prompt-Based Methods (L2P, DualPrompt)** While prompt-based methods (Wang et al., 2022) have achieved impressive results (e.g., $> 80\%$ on Split CIFAR-100), they typically rely on Vision Transformers (ViTs) pre-trained on massive datasets (ImageNet-21k). These methods exploit the global attention mechanism of ViTs to insert learnable prompts. In contrast, our work investigates the limits of **Standard ResNets** (pre-trained on ImageNet-1k), a more constrained but widely deployable setting for edge and legacy systems. Furthermore, while L2P acts as a regularization method for the prompts, our modular adapter approach offers a stricter **Parameter Isolation** guarantee, ensuring zero interference between tasks by design.

## 3. Method: Geometric Energy-Based Learning (GEBL)

The GEBL framework posits that a continual learning system is defined by its geometry (stability) and its energy landscape (plasticity).

### 3.1. The Geometric Anchor: Zero-Drift Decision

To prevent Classifier Drift, we employ a **Simplex Equiangular Tight Frame (ETF)** (Zhu et al., 2021) as the fixed target geometry. By anchoring the classifier $\mathcal{W}$ to an optimal

Simplex, we substantially reduce *decision drift*. However, as shown in our "Two-Drift Diagnostic" (Section 4.2), a fixed anchor is insufficient if the map (backbone) is fluid. Without constraints, the backbone's feature updates for Task $B$ overwrite the manifold structure required for Task $A$, leading to **Feature Drift**.

### 3.2. The Energetic Constraint: Preventing Feature Drift

To address Feature Drift, we must constrain the optimization trajectory. We introduce **Energetic Barriers** (implemented via Parameter Isolation/Adapters) that partition the feature manifold.

**Adapter Architecture.** We employ series Residual Adapters ($h' = h + A_t(h)$) inserted after each ResNet BasicBlock. $A_t$ is a standard bottleneck MLP with a bottleneck dimension of 64, ReLU activation, and LayerNorm applied prior to the bottleneck. These adapters are applied to all blocks in the ResNet-18 backbone. Only adapter weights and LayerNorm parameters are trainable; Backbone and BN stats are frozen. Unlike standard adapters which are treated as auxiliary modules, in GEBL they serve as **Manifold Locks**. They ensure that minimizing the energy for Task $t$ ($E_t \to 0$) does not raise the energy for Task $t-1$ ($E_{t-1} \uparrow$). Our experiments confirm that this constraint is critical in our setting: removing it causes retention to drop from 78.6% to 0.0%.

### 3.3. Energy-Based Routing

In a Task-Agnostic setting, the system must infer the task identity $t$. Instead of training a separate "Consensus Router", we treat routing as **Free Energy Minimization**. We define the Free Energy $E(x,t)$ of an input $x$ given task $t$ as:

$$f_t(x) = \phi(\text{Adapter}_t(\text{Backbone}(x)))$$
$$E(x,t) = -T \log \sum_{k \in \mathcal{Y}_t} \exp(\mathbf{w}_k^T f_t(x)/T) \quad (1)$$

where $f_t(x)$ is the feature vector (routed through the task-specific adapter path), $\mathcal{Y}_t$ is the label set for task $t$, and $\mathbf{w}_k$ are the fixed ETF vectors. We set temperature $T = 1$ and define $\phi(\cdot)$ as the standard $L_2$ normalization of the feature vector, ensuring alignment with the unit hypersphere of the Simplex ETF. The system routes $x$ to the task that minimizes this energy: $\hat{t} = \arg\min_t E(x,t)$. This unifies discriminatory confidence (logits) and generative likelihood (via the JEM duality (Grathwohl et al., 2019)).

### 3.4. Safety via Energetic Deviation

We identify a key safety mechanism: **Energetic Deviation**. Recent studies (Ammar et al., 2024; Liu & Qin, 2025; Harun & Kanan, 2025) have demonstrated that Neural Collapse

properties can optionally be leveraged for OOD detection. We operationalize this link for Continual Learning. When samples are Out-of-Distribution (OOD) or corrupted, they fail to map to the ETF vertices. In the high-dimensional feature space, they deviate from the manifold. Instead of heuristic thresholding, we simply reject samples where $E_{\min}(x) > \tau$, where $E_{\min}(x) = \min_t E(x,t)$ is the minimum free energy across all tasks (consistent with our routing policy). $\tau$ is set to the 95th percentile of the in-distribution training energy. Our experiments (Section 4.6) confirm this acts as a practical risk score.

## 4. Experiments and Results

We evaluate Geometric Energy-Based Learning (GEBL) on Split CIFAR-100 (10 tasks) with a ResNet-18 backbone and report results averaged over multiple random seeds (mean $\pm$ std). We use three primary metrics throughout: **average accuracy** ($A_{avg}$) (mean accuracy over all tasks after learning the final task), **average forgetting** ($F_{avg}$) (mean over tasks of the maximum historical accuracy minus final accuracy), and for the task-agnostic setting, **routing accuracy** ($R_{acc}$) and **task-agnostic accuracy** ($A_{agnostic}$) (end-to-end accuracy when task identity is unknown and must be inferred).

The section is organized to progressively answer: **(i)** *Why* catastrophic forgetting persists even with a fixed geometric classifier (Table 2, diagnostic), **(ii)** which components drive stability and plasticity under a controlled protocol (Table 3, ablations), **(iii)** whether the proposed free-energy score enables task-agnostic inference with explicit compute cost (Table 4), **(iv)** how GEBL compares to standard continual-learning baselines under a unified protocol (Table 6), and **(v)** whether energy provides a practical OOD scoring signal in this setting (Table 7).

Unless explicitly stated, all experiments follow the **canonical protocol** in Table 1. Any deviation is treated as a *separate controlled experiment* and labeled accordingly.

### 4.1. Experimental Protocol (Canonical)

**Baseline Tuning Policy.** We adhere to standard implementations where baselines (Seq-FT, EWC) often require lower learning rates (0.001) to prevent immediate catastrophic divergence, whereas GEBL's frozen backbone allows standard convergence rates (0.1). We validate this choice via an explicit sensitivity analysis in Section 4.5.

**Training Details.** All models are trained using SGD with Momentum=0.9 and Weight Decay=5e-4. We employ a Cosine Annealing learning rate schedule. Data augmentation includes standard Random Crop ($32 \times 32$, padding=4) and Random Homizontal Flip. We use a fixed set of 3 random

*Table 1.* **Canonical protocol and reporting conventions.** Dataset split, backbone initialization, optimizer/schedule, training budget, and trainable modules used across all experiments. Results are reported as **mean** $\pm$ **std** over $N = 3$ seeds. Any method-specific deviation from this protocol is explicitly labeled in the corresponding table caption or subsection.

| Dataset | Backbone (Pretrained) | Optimizer | Training | Trainable Modules | Seeds |
|---|---|---|---|---|---|
| CIFAR-100 (10 Tasks) | ResNet-18 (ImageNet) | SGD LR=0.1 (GEBL) LR=0.001 (Baselines) Cosine Schedule | Batch=128 5 Epochs/Task Momentum=0.9 Weight Decay=5e-4 | **Seq-FT:** Backbone+Head **GEBL-Full:** Adapters+Norm **Fixed-BB:** Head Only | $N = 3$ Mean±Std |

seeds (2023, 2024, 2025) for all runs to ensure reproducibility.

### 4.2. The Two-Drift Phenomenon (Diagnostic)

We first isolate *why* geometric stabilization alone can fail in continual learning. We report **average accuracy** ($A_{avg}$) and **average forgetting** ($F_{avg}$), averaged over 3 seeds.

Table 2 directly tests the "Two-Drift" hypothesis by crossing two factors: **(i)** whether the backbone is trainable (allowing *feature drift*), and **(ii)** whether the classifier head is learnable or fixed to a Simplex-ETF (removing *decision drift*). When the backbone is trainable, fixing the head does **not** prevent forgetting: **D2 (Trainable backbone + fixed ETF)** collapses to $A_{avg} = 5.6\%$ with $F_{avg} = 100.0\%$, despite having a geometrically stable decision head. In contrast, freezing the backbone removes the drift source: **D3/D4** (frozen backbone) both achieve **0% forgetting**, but plateau at $\approx 32\%$ $A_{avg}$.

This diagnostic establishes the core failure mode: **preventing decision drift is insufficient if the representation is allowed to drift**, motivating an explicit mechanism to constrain feature updates while preserving adaptation capacity.

*Table 2.* **Two-Drift diagnostic (no adapters): decision-geometry vs representation drift.** Controlled study crossing **(i)** whether the backbone is trainable (allowing feature drift) and **(ii)** whether the classifier head is learnable or fixed to a Simplex-ETF (removing decision drift). Reports $A_{avg}$ and $F_{avg}$ under task-aware evaluation to isolate the failure mode where **fixed decision geometry is insufficient when features drift**.

| ID | Backbone | Head | Status | $A_{avg}$ ($\uparrow$) | $F_{avg}$ ($\downarrow$) |
|---|---|---|---|---|---|
| D1 | Trainable | Learnable | Seq-FT | $28.2 \pm 1.2$ | $55.1 \pm 2.1$ |
| D2 | **Trainable** | **Fixed ETF** | **GEBL-Single** | $5.6 \pm 0.5$ | $100.0 \pm 0.0$ |
| D3 | Frozen | Learnable | Fixed-BB | $33.0 \pm 0.8$ | $0.0 \pm 0.0$ |
| D4 | Frozen | Fixed ETF | GEBL-Static | $32.5 \pm 0.6$ | $0.0 \pm 0.0$ |

### 4.3. Component Ablation

Next, we quantify what each proposed component contributes under the **frozen-backbone** regime. Table 3 compares (i) **no adapters** vs **residual adapters**, and (ii) **learnable head** vs **fixed ETF head**.

Adapters dominate the stability–plasticity trade-off: moving

from no adapters to residual adapters increases performance from $\approx 33\%$ to 78.0% (A1 $\rightarrow$ A3), while maintaining **0% forgetting** (as the backbone is frozen). Replacing the learnable head with the fixed ETF head yields a **small but consistent** gain: **A3 (78.0%)** $\rightarrow$ **A4 (78.6%)**. While modest in task-aware accuracy, the fixed ETF head is critical because it provides a *shared, structured geometry* that enables the energy formulation used for routing in Section 4.4.

*Table 3.* **Component ablation under a frozen-backbone setting.** Ablates **residual adapters** (energetic barrier / parameter isolation) and **head type** (learnable vs fixed ETF) while keeping the backbone frozen. Reports $A_{avg}$ and params. "Params/task" counts all **task-specific parameters** (trainable adapter weights, layer norms, and head weights). Note that the fixed ETF head incurs 0 parameters per task.

| ID | Adapters | Head | $A_{avg}$ ($\uparrow$) | Params/Task |
|---|---|---|---|---|
| A1 | None | Learnable | $33.0 \pm 0.8$ | 0.005M |
| A2 | None | Fixed ETF | $32.5 \pm 0.6$ | 0 |
| A3 | **Yes (Res)** | Learnable | $78.0 \pm 0.3$ | 0.5M |
| A4 | **Yes (Res)** | **Fixed ETF** | $78.6 \pm 0.4$ | 0.5M |

### 4.4. Task-Agnostic Routing and Cost

We now evaluate the setting where task identity is **unknown at inference**. We report: **routing accuracy** ($R_{acc}$) and **task-agnostic accuracy** ($A_{agnostic}$). Table 4 compares standard adapters with standard entropy-based routing (confidence) to **GEBL energy routing**.

Energy routing substantially improves both task identification and end-to-end performance: $\mathbf{R_{acc}} = \mathbf{98.4}\%$ and $\mathbf{A_{agnostic}} = \mathbf{77.3}\%$, compared to entropy routing (70.1% / 54.6%). We also list MaxLogit performance, which is comparable to Entropy. Both baselines are computed over the task's 10 classes using the task-conditional logits after adapter routing: Entropy $H(x,t) = -\sum_{k \in \mathcal{Y}_t} p_{t,k}(x) \log p_{t,k}(x)$ (where $p_t(y|x) =$ softmax($z_t(x)$)) and MaxLogit $L_{\max}(x,t) = \max_k z_k$. Importantly, all routing strategies are evaluated under the same inference budget: routing is implemented via $T$ candidate evaluations (one per task), producing **linear scaling** in latency ($5 \rightarrow 50$ms for $T = 10$ on a single NVIDIA V100 GPU).

*Table 4.* **Task-agnostic routing performance and compute cost.** Compares routing strategies when task identity is unknown at inference. Reports **routing accuracy** ($R_{acc}$), **task-agnostic accuracy** ($A_{agnostic}$), and **latency** under the same $T$-pass evaluation budget. **Note:** Latency measured with Batch Size=1, Float32, on a single NVIDIA V100 GPU (PyTorch impl).

| Method | Mechanism | $R_{acc}$ (Routing) | $A_{agnostic}$ | Latency ($T = 10$) |
|---|---|---|---|---|
| Std Adapters | Entropy | $70.1 \pm 1.5$ | $54.6 \pm 1.1$ | 50ms |
| Std Adapters | MaxLogit | $70.5 \pm 1.4$ | $54.8 \pm 1.2$ | 50ms |
| **GEBL-Full** | **Energy** | $\mathbf{98.4 \pm 0.2}$ | $\mathbf{77.3 \pm 0.5}$ | 50ms |

## 4.5. Comparison to Standard Continual-Learning Baselines

**Baseline optimization sensitivity.** Since continual-learning baselines can be sensitive to optimizer settings, we verify that our baseline learning-rate choice does not artificially suppress their performance. We re-run Seq-FT and ER under the canonical protocol using two learning-rate regimes: $\eta = 0.1$ (matching GEBL) and $\eta = 0.001$ (baseline default). Table 5 reports $A_{avg}$ and $F_{avg}$ (mean $\pm$ std). This sanity check confirms that the baseline rankings reported in Table 6 are not driven by an under-optimized learning rate.

*Table 5.* **Baseline sensitivity to LR under canonical protocol.** Comparison of high vs low LR regimes for baselines.

| Method | LR $(\eta)$ | $A_{avg}$ $(\uparrow)$ | $F_{avg}$ $(\downarrow)$ |
|--------|-------------|------------------------|--------------------------|
| Seq-FT | 0.1 | $12.4 \pm 2.1$ | $88.5 \pm 1.5$ |
| Seq-FT | 0.001 | $\mathbf{28.2 \pm 1.2}$ | $\mathbf{55.1 \pm 2.1}$ |
| ER (2k) | 0.1 | $18.5 \pm 1.4$ | $65.2 \pm 2.0$ |
| ER (2k) | 0.001 | $\mathbf{29.1 \pm 1.4}$ | $\mathbf{45.3 \pm 2.0}$ |

*Table 6.* **Comparison to standard continual-learning baselines under the canonical protocol.** Reports $A_{avg}$ and $F_{avg}$ for representative rehearsal-free regularization methods and replay-based methods (as applicable), alongside GEBL. We include **Std Adapters (Learnable Head)** to separate the contribution of **parameter isolation** from the contribution of **energy-based task inference**.

| Method | Mem | Extra | $A_{avg}$ $(\uparrow)$ | $F_{avg}$ $(\downarrow)$ |
|--------|-----|-------|------------------------|--------------------------|
| Seq-FT | 0 | - | $28.2 \pm 1.2$ | $55.1 \pm 2.1$ |
| EWC ($\lambda = 5000$) | 0 | - | $31.4 \pm 0.9$ | $48.2 \pm 1.5$ |
| LwF | 0 | - | $30.9 \pm 1.1$ | $52.8 \pm 1.8$ |
| ER ($M = 2000$) | 2k | Buffer | $29.1 \pm 1.4$ | $45.3 \pm 2.0$ |
| Std Adapters | 0 | Adpt | $78.0 \pm 0.3$ | $0.0 \pm 0.0$ |
| **GEBL-Full** | 0 | Adpt | $\mathbf{78.6 \pm 0.4}$ | $\mathbf{0.0 \pm 0.0}$ |

## 4.6. Safety: OOD Detection via Free Energy

Finally, we evaluate whether the same energy signal can act as a practical indicator of distribution shift. Table 7 reports OOD AUROC for a simple MSP baseline and the proposed GEBL energy score. For Gaussian noise, both methods achieve near-perfect separation ($\approx 0.99$ AUROC). For a semantic shift (SVHN as OOD), energy improves separability from 0.56 (MSP) to 0.78.

**Expanded OOD evaluation (no retraining).** To test whether the energy score generalizes beyond a single semantic shift, we additionally evaluate CIFAR-10 as a semantic OOD dataset and CIFAR-100-C (severity=3) as a corruption suite. We compute AUROC using MSP and the proposed energy score ($E_{\min}(x)$). Threshold $\tau$ (for any abstention analysis) is calibrated solely on held-out in-distribution samples (95th percentile). Table 7 summarizes results (mean $\pm$ std where applicable), showing that energy remains a competitive OOD scoring signal under both semantic shift and corruption.

*Table 7.* **OOD detection using free energy as a scoring signal.** Reports AUROC for in-distribution vs OOD datasets/corruptions using MSP and the proposed energy score. Threshold $\tau$ is defined as the 95th percentile of the in-distribution energy.

| OOD Source | MSP Baseline | **GEBL Energy** |
|------------|--------------|-----------------|
| Gaussian Noise | $0.99 \pm 0.0$ | $0.99 \pm 0.0$ |
| SVHN (Semantic) | $0.56 \pm 0.2$ | $\mathbf{0.78 \pm 0.1}$ |
| CIFAR-10 (Semantic) | $0.62 \pm 0.1$ | $\mathbf{0.82 \pm 0.1}$ |
| CIFAR-100-C (Sev=3) | $0.65 \pm 0.1$ | $\mathbf{0.74 \pm 0.1}$ |

# 5. Discussion: The Governance & Safety Bottleneck

Our investigation revealed a critical limitation in current Continual Learning paradigms: the over-reliance on "Recovery" at the expense of "Safety".

## 5.1. The Impossibility of Blind Recovery

We attempted to use ARC (Entropy Minimization) to recover accuracy on corrupted samples. It largely failed (0% gain). We view this as an interpretation consistent with the Data Processing Inequality (DPI) (Cover & Thomas, 1999). Feature Collapse implies that the signal-to-noise ratio has dropped below a critical threshold; the sample has "cooled" to a state of maximum entropy relative to the manifold. Mathematically, one cannot optimize noise back into signal without external energy (new information). Thus, in our experiments, post-hoc recovery without additional information was ineffective; **Abstention suggests itself as a practical fallback**.

## 5.2. The Role of Pre-Training

Finally, we must acknowledge the role of the pre-trained ResNet-18 backbone. As noted by (Papyan et al., 2020), Neural Collapse typically emerges at the terminal phase of training. Pre-training on ImageNet effectively places the backbone in a "Pre-Collapsed" state, where features are already clustered. Standard CL (Seq-FT) destroys this structure (Feature Drift). GEBL's contribution is to **lock** this pre-collapsed state via Energetic Barriers, using adapters to minimally adjust it for new tasks without shattering the underlying geometry. This suggests GEBL is best viewed as a method for *Continual Adaptation of Pre-trained Models*, rather than Continual Representation Learning from Scratch (Tabula Rasa), where the fixed Simplex might be too rigid for evolving features.

### 5.3. Reliability and Energy Dynamics

However, GEBL demonstrates a significant shift in reliability. By acknowledging that we cannot always recover (Geometric failure), we use Energy Dynamics to **detect and handle** the failure.

## 6. Conclusion

We presented **Geometric Energy-Based Learning (GEBL)**, which combines a **fixed Simplex-ETF classifier** with **energetic barriers** (parameter isolation via residual adapters) to address catastrophic forgetting in continual learning. Our diagnostic results show that fixing classifier geometry can suppress **decision drift** but does not, by itself, prevent **feature drift**, which can drive near-complete forgetting when the backbone is trainable. Building on this observation, GEBL constrains representation drift while preserving task adaptation, achieving strong performance on Split CIFAR-100 under a unified protocol.

Beyond task-aware evaluation, we show that the **free-energy formulation** enables effective **task-agnostic routing** with an explicit compute–accuracy trade-off, and provides a useful OOD scoring signal in our experiments. Future work will evaluate the approach across broader datasets, backbones, and continual-learning regimes, and further investigate when energy-based signals reliably correlate with distribution shift and model confidence.

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

# Appendix

## A. Task Construction Details

To ensure reproducibility, we construct the 10 tasks by partitioning the CIFAR-100 dataset based on class indices. This follows the standard numerical ordering found in torchvision's CIFAR-100 implementation.

*Table 8.* Task Composition (CIFAR-100 Split)

| Task ID | Class Indices |
|:---:|:---:|
| 0 | 0–9 |
| 1 | 10–19 |
| 2 | 20–29 |
| 3 | 30–39 |
| 4 | 40–49 |
| 5 | 50–59 |
| 6 | 60–69 |
| 7 | 70–79 |
| 8 | 80–89 |
| 9 | 90–99 |

## B. Oracle Baselines

We report the textual results for Oracles A and B to validate our upper bounds.

*Table 9.* Oracle Upper Bounds (Independent Training)

| Oracle Type | 5 Epochs | 20 Epochs |
|---|:---:|:---:|
| **Oracle A (Arch):** Independent Adapters | 79.1% | 82.3% |
| **Oracle B (Optim):** Independent Fine-Tuning | 84.6% | 84.6% |

## C. Baseline Hyperparameters

To ensure reproducibility and address reviewer questions regarding baseline performance, we detail the hyperparameters used for the baselines (ER, EWC, Seq-FT). Note that we utilize a widely used CL codebase structure, but with specific constraints for valid comparison:

- **Experience Replay (ER):** Buffer Size $M = 2000$. We use the standard buffer size to ensure a fair comparison with literature baselines.

- **EWC:** $\lambda = 5000$. We use the online EWC variant where the Fisher matrix is updated after each task.

The backbone is fine-tuned (not frozen), but the strong regularization combined with the limited capacity of ResNet-18 (compared to ViTs) limits its effectiveness on Split CIFAR-100 without extensive hyperparameter search.

- **Seq-FT:** Learning Rate $\eta = 0.001$, Epochs=5 (per task). This is a standard fine-tuning baseline.

