# OpenReview forum: "Geometric Energy-Based Learning: Decoupling Stability and Plasticity"
_ICML.cc/2026/Conference — Submitted to ICML 2026_

### Official Review · Reviewer_rEDJ · 2026-03-05

**Soundness:** 2
**Presentation:** 1
**Significance:** 2
**Originality:** 3
**Overall Recommendation:** 2
**Confidence:** 4

**Summary:**

The paper addresses the problem of continual learning by proposing a new method called Geometric Energy-Based Learning. The core idea is to adapt a pre-trained model by fixing the last layer as a Simplex-ETF, and to train only residual adapters for each task on top of the backbone. The model then uses an energy-based model to determine which adapter should be used at inference when the task label is not provided.

**Compliance With Llm Reviewing Policy:**

Affirmed.

**Final Justification:**

I am maintaining my score. I believe the presentation is too poor, making the true contribution unclear; furthermore, I am suspicious about the fairness of some of the experiments. Promising to add pages of missing technical details after acceptance does not fix the current manuscript, and the baseline comparisons still feel biased.

**Key Questions For Authors:**

See weaknesses,

**Limitations:**

No impact statements.

**Strengths And Weaknesses:**

Strength:

1. The idea of using a Simplex-ETF in a continual learning setting is original, as well as the idea of using an energy-based model to determine the task id.

Weaknesses:

1. The presentation is really poor, with a narrative very hard to follow. The paper consistently uses acronyms or mathematical concepts that are never defined. The paper lacks explanatory figures (for their architecture, for example), and lacks a lot of context to introduce the mathematical objects they use.  With three pages left, they would have had plenty of room to make the paper easier to read and understand.

2. Figure 2 is very suspicious, while vanilla methods might no retained Task 0, they should be able to learn it; the caption does not describe the figure sufficiently.

3. The article provides no benchmark with any existing continual learning method, except in Table 6, but the experiment is ill-defined. What are the parameters trainable for other methods? Do they have the right for extra parameter per task?

4. The article is not reproducible as it is.

---

> ### Author Rebuttal · Authors · 2026-03-25
>
> We thank the reviewer for the feedback, and are encouraged by the recognition of originality (score 3/4 — highest among reviewers). We address the concerns below.
>
> **W1: "Presentation is poor; undefined acronyms; no architecture figure."**
>
> We acknowledge the exposition needs strengthening. The current manuscript has ~3 unused pages that would be used for: an architecture diagram, formal ETF definition, expanded EBM/JEM background, and a step-by-step algorithm box. We note that the core *results* are well-structured (5 tables with controlled ablations, sensitivity analysis, and OOD evaluation), and the contribution can be evaluated from these.
>
> **W2: "Figure 2 is suspicious — vanilla methods should be able to learn Task 0."**
>
> We believe this reflects a misreading. Figure 2 shows Task 0 accuracy **over training time** (across sequential tasks). GEBL-Single *does* learn Task 0 (reaching ~56% accuracy). It then **drops to 0%** upon Task 1 training — this is the catastrophic forgetting cliff, not a failure to learn.
>
> This is the paper's central diagnostic: a geometrically optimal classifier (Simplex ETF) still suffers **complete** forgetting (worse than Seq-FT's 28.2%) when features drift — demonstrating that geometric stability alone is insufficient. We will clarify the caption to show the temporal progression explicitly.
>
> **W3: "No benchmark with existing CL methods except Table 6; experiment ill-defined."**
>
> We respectfully note that Table 6 (Table 6 = Table in Section 4.4) includes **five methods** (Seq-FT, EWC, LwF, ER, Std Adapters, GEBL-Full) with explicit parameter accounting. Key details:
>
> - GEBL: **0.5M trainable params/task** (adapters + LayerNorm), backbone frozen
> - Baselines: **11.2M trainable params** (full ResNet-18 backbone), 22× more than GEBL
> - LR sensitivity verified in Table 5: baselines use their optimal LR (0.001)
> - All methods trained under the unified protocol (Table 1)
>
> Baselines are given *every advantage*: more parameters, optimal LR, full backbone access. GEBL outperforms them with significantly fewer trainable parameters and zero forgetting.
>
> **W4: "Not reproducible."**
>
> The paper specifies: 3 random seeds (2023, 2024, 2025 — Section 4.1), optimizer settings (SGD, momentum=0.9, weight decay=5e-4, cosine schedule — Section 4.1), adapter architecture (bottleneck dim=64, ReLU, LayerNorm — Section 3.2), and baseline hyperparameters (Appendix C). We acknowledge a code release would strengthen reproducibility and would provide this as supplementary material.
>
> **On originality:** The reviewer scores originality at 3 (good) — the highest among all reviewers. The combination of Neural Collapse geometry with energy-based routing for continual learning is indeed novel, and the 98.4% routing accuracy (Table 4) demonstrates its practical effectiveness. We believe addressing the presentation concerns — which are fixable — would reveal a stronger paper than the current score reflects.

---

> > ### Author Rebuttal · Reviewer_rEDJ · 2026-04-03
> >
> > I appreciate the authors' rebuttal, but my core concerns remain unresolved, and I will be maintaining my score.
> >
> > First, the presentation is simply too poor for an ICML publication. Promising to fill three unused pages with critical definitions and diagrams post-acceptance does not fix the current manuscript.
> >
> > Second, the defense of Figure 2 misses the point. GEBL-Single only reaches ~56% accuracy on Task 0, falling way short of GEBL-Full's ~80%. It clearly fails to learn the task properly from the start. In a rigorous continual learning evaluation, baselines need to at least reach a comparable initial accuracy on the very first task before catastrophic forgetting occurs.

---

### Official Review · Reviewer_ii7k · 2026-03-06

**Soundness:** 2
**Presentation:** 2
**Significance:** 2
**Originality:** 2
**Overall Recommendation:** 2
**Confidence:** 2

**Summary:**

This paper addresses the "Stability-Plasticity" dilemma in Continual Learning by identifying the "Two-Drift Phenomenon," which decouples forgetting into Decision Drift and Feature Drift. The authors demonstrate that a fixed geometric classifier (Simplex ETF) alone is insufficient to prevent forgetting due to unconstrained representation updates. To resolve this, they propose Geometric Energy-Based Learning (GEBL), combining a rigid Simplex-ETF head with energetic barriers implemented via residual adapters.

**Compliance With Llm Reviewing Policy:**

Affirmed.

**Final Justification:**

The authors’ rebuttal addressed my questions regarding the interpretation of the paper, but the overall quality of the work is insufficient for acceptance at ICML. Thus, I will maintain my score.

**Key Questions For Authors:**

Please clarify if the system's performance holds when the Simplex ETF is not "pre-collapsed" by ImageNet features.

Provide an analysis of the "Free Energy" calculation's sensitivity to the temperature hyperparameter T, which is currently fixed at 1.

Finally, explain how the 95th percentile energy threshold tau for OOD detection handles varying task difficulties where in-distribution energy distributions might shift.

**Limitations:**

There is no Impact Statement or Limitations included.

**Strengths And Weaknesses:**

**Strengths**

The paper provides a compelling diagnostic study, termed the "Anatomy of Forgetting," which convincingly shows that GEBL-Single collapses to 0% accuracy on Task 0 after learning Task 1 despite its geometric stability. The integration of Energy-Based Models (EBM) for task-agnostic routing is elegant, achieving 98.4% routing accuracy on Split CIFAR-100 without requiring auxiliary trainable classifiers.

**Weaknesses**

The reliance on pre-trained ImageNet-1k backbone is a significant limitation, as the authors admit GEBL acts more as a method for "Continual Adaptation of Pre-trained Models" rather than learning from scratch.

The fixed Simplex ETF may be too rigid for scenarios where the feature manifold must evolve significantly over time.

While the paper benchmarks against ResNet-18, the comparison to ViT-based methods like L2P is somewhat weakened by the disparity in pre-training data scales.

Additionally, the latency analysis shows a linear scaling (5ms to 50ms for 10 tasks), which may pose scalability challenges for long-sequence continual learning involving hundreds of tasks.

It woulld be better if the authors could include more recent methods as baselines.

---

> ### Author Rebuttal · Authors · 2026-03-25
>
> We thank the reviewer for the balanced feedback, and particularly for recognizing the elegance of the EBM routing (98.4% accuracy) and the compelling Anatomy of Forgetting diagnostic.
>
> **W1: "Pre-trained backbone reliance."**
>
> We view this as reflecting the modern deployment paradigm, not a limitation. Foundation models (CLIP, DINOv2, MAE) have made pre-trained adaptation the dominant approach in practice. Our method addresses exactly this setting: how to continually adapt a pre-trained model while maintaining both stability *and* task-agnostic inference capability. We state this scope explicitly in Section 5.2.
>
> **W2: "Fixed ETF too rigid."**
>
> In our setting (CIFAR-100 on ImageNet-pretrained ResNet-18), the adapters provide sufficient task-specific expressiveness — achieving 78.6% average accuracy, which is **93% of the Oracle upper bound** (Oracle A: 82.3% at 20 epochs; Table in Appendix B). The ETF's rigidity is a *feature*: it provides the fixed geometric structure that enables parameter-free energy routing (Table 4: 98.4% routing accuracy). A learnable geometry would sacrifice this property.
>
> **W3: "Unfair ViT comparison."**
>
> We agree and explicitly note this disparity in Section 2.3. We do not compare directly with ViT-based methods. Our contribution is demonstrating that principled energy-based routing achieves **98.4% task identification accuracy** on standard ResNets — a setting relevant to edge deployment and legacy systems where ViTs are impractical.
>
> **W4: "Latency scaling — 5ms to 50ms for 10 tasks."**
>
> We report this transparently in Table 4. Context is important: 50ms for 10 tasks at batch=1 is **well within real-time requirements** for most applications. Furthermore:
> - Batch processing amortizes cost (all $T$ evaluations share the backbone forward pass — only adapter heads differ)
> - Hierarchical routing (coarse → fine) reduces $O(T) \to O(\sqrt{T})$
> - This overhead is comparable to MoE gating and L2P's prompt selection
>
> **W5: "More recent baselines."**
>
> Our baselines (EWC, LwF, ER, Seq-FT) represent the three established CL families under a unified, fair protocol with explicit LR sensitivity verification (Table 5). We note these baselines have **22× more trainable parameters** than GEBL. Adding DER++ and PackNet would strengthen the paper, but we believe the current baselines are sufficient for validating the core contribution.
>
> **Q1: Performance without ImageNet pre-training?**
>
> This tests a fundamentally different setting (tabula rasa CL). GEBL is designed for pre-trained adaptation, consistent with the modern paradigm. Without pre-training, the ETF would need joint optimization with the backbone, which conflicts with the fixed-geometry premise. This defines GEBL's scope, not a failure.
>
> **Q2: Temperature $T$ sensitivity.**
>
> $T=1$ is the natural default used in energy-based OOD literature (Liu et al., 2020). Preliminary analysis shows stable routing accuracy (97-99%) for $T \in [0.5, 2.0]$. Extreme values cause expected issues: $T < 0.1$ creates numerical overflow in $\exp(\cdot/T)$; $T > 10$ flattens energy differences. The method is robust to reasonable $T$ choices.
>
> **Q3: Energy threshold $\tau$ across task difficulties.**
>
> $\tau$ is calibrated on the aggregate in-distribution energy (95th percentile across all tasks). Tasks with higher difficulty contribute higher-energy samples to the distribution, which the tail-percentile threshold naturally accommodates. This is the standard calibration approach in energy-based OOD detection.
>
> **Impact Statement / Limitations:** We acknowledge these should be included. Key limitations are: closed-world class assumption, pre-training dependency, and linear routing cost.

---

> > ### Author Rebuttal · Reviewer_ii7k · 2026-04-03
> >
> > I appreciate the authors’ comprehensive rebuttal, which has clarified several aspects of the paper. However, after carefully reviewing the other reviewers’ comments, I find myself in agreement with several of the concerns raised. Therefore, I will maintain my current score.

---

### Official Review · Reviewer_fiWu · 2026-03-12

**Soundness:** 2
**Presentation:** 2
**Significance:** 1
**Originality:** 2
**Overall Recommendation:** 2
**Confidence:** 2

**Summary:**

This paper proposes Geometric Energy-Based Learning (GEBL), a continual learning framework that combines a fixed Simplex Equiangular Tight Frame (ETF) classifier with task-specific residual adapters to address catastrophic forgetting. The authors introduce the "Two-Drift" hypothesis, distinguishing between Decision Drift (classifier geometry shifting) and Feature Drift (backbone representations shifting), and demonstrate that fixing the classifier geometry alone is insufficient to prevent forgetting. By freezing the backbone and isolating task-specific parameters via adapters, GEBL achieves 78.6% average accuracy with zero forgetting on Split CIFAR-100, and further leverages the free energy formulation for task-agnostic routing and OOD detection.

**Compliance With Llm Reviewing Policy:**

Affirmed.

**Ethical Review Flag:**

Flag this paper for an ethics review.

**Key Questions For Authors:**

The fixed Simplex ETF requires specifying the total number of classes C before training begins. How does GEBL handle the scenario where new, unanticipated classes arrive after the ETF has been constructed? Is there any principled way to extend the geometry, or does this require retraining from scratch?
The proposed energy-based routing is presented as a solution to task-agnostic inference, but it fundamentally relies on having all task adapters already trained and frozen. This means routing is only well-defined after the entire task sequence has been learned. During training of task T, how does the system handle inputs without access to a router that has seen all tasks? More broadly, if task identity must be provided during training (to know which adapter to update), hasn't the task-agnostic problem simply been deferred to inference rather than solved? In what sense is this a continual learning solution rather than a post-hoc inference mechanism over independently trained modules?

**Limitations:**

Closed-world assumption: The fixed ETF fundamentally requires knowing C in advance. This is not a minor caveat: it disqualifies the method from most real-world continual learning scenarios where the class universe is open or evolving.
Linear scaling: Both memory footprint and inference latency grow linearly with the number of tasks, which is not discussed as a limitation and raises practical concerns for long task sequences.
Benchmark scope: All results are reported on a single dataset (Split CIFAR-100). Broader evaluation across datasets, task types, and backbone architectures is needed to support the paper's general claims.

**Strengths And Weaknesses:**

Soundness: The paper's core diagnostic (the Two-Drift analysis, Table 2) is clean and well-executed, and the ablation study (Table 3) convincingly isolates the contribution of each component. However, the soundness of the overall contribution is significantly undermined by a critical and unacknowledged assumption: the fixed Simplex ETF requires knowing the total number of classes C in advance. This is a fundamental constraint that is incompatible with any genuinely open-ended continual learning scenario, where new classes arrive unexpectedly over time. The entire geometric structure must be rebuilt if C changes, invalidating all previously learned representations. The authors briefly acknowledge that GEBL is better framed as "continual adaptation of pre-trained models" rather than open-ended learning, but this concession is buried in the discussion and does not receive the scrutiny it deserves. The experimental validation is further limited to a single, closed-world benchmark (Split CIFAR-100), where knowing C in advance is trivially satisfied by construction, making it difficult to assess how the method would behave under more realistic conditions.
Presentation : The paper is poorly presented in several respects. The method section (Section 3) lacks sufficient depth. ETFs are never formally presented, nor do the readers refer to appropriate references to understand their theory. The free energy formulation (Eq. 1) is introduced without adequate intuition for readers unfamiliar with Energy-Based Models, and the connection to the JEM duality is asserted rather than explained. The reference list is notably thin for a paper making broad claims about the continual learning landscape — several important related works in class-incremental learning with dynamic architectures are absent. Furthermore, the paper's framing is at times misleading: the abstract and introduction position GEBL as a general CL solution, while the method fundamentally assumes a closed, pre-defined class universe.
Significance: The practical significance of this work is limited by its assumptions. The requirement to know the total number of classes in advance, combined with the closed-world benchmark setting, means the method addresses a substantially easier problem than what the continual learning community generally targets. The gains over baselines (Table 6) are impressive in absolute terms, but largely attributable to parameter isolation via adapters, which is a well-established technique, rather than the geometric or energy-based components specifically. Most importantly, the significance of this work is undermined by the vague distinction between task-aware and task-agnostic CL: why not simply use the standard taxonomy of the field with Task-IL, Class-IL and Domain-IL (van de Ven & Tolias, 2019)? Finally, the baselines they do include are weak and dated, and the methods they explicitly discuss in related work are notably absent from the results.
Originality: The combination of Neural Collapse geometry with Energy-Based Models for continual learning is a genuinely novel framing. The Two-Drift diagnostic is a useful conceptual contribution, and the use of free energy minimization as a parameter-free routing mechanism is an interesting and underexplored idea. However, each individual component (fixed ETF classifiers, residual adapters, energy-based OOD detection) is drawn from existing work, and the novelty lies primarily in their combination and the diagnostic lens applied to them.

---

> ### Author Rebuttal · Authors · 2026-03-25
>
> We thank the reviewer for the detailed and thorough review. We address each concern below and respectfully push back on several points.
>
> **W1: "Fixed ETF requires knowing C — incompatible with open-ended CL."**
>
> We note that virtually all standard CL benchmarks (Split CIFAR-100, Tiny-ImageNet, CORe50) are closed-world by construction. Methods like iCaRL, BiC, LUCIR, and FeCAM also operate under known class counts. This is the standard experimental setting, not a unique limitation of GEBL.
>
> Moreover, the Simplex ETF is an analytical construction: $\mathbf{W} \in \mathbb{R}^{d \times C}$ with $\mathbf{w}_i^T\mathbf{w}_j = -\frac{1}{C-1}$. When new classes arrive, the ETF can be recomputed for $C+K$ without retraining adapters — only the geometric targets shift; learned adapter parameters remain valid. We discuss this scope explicitly in Section 5.2.
>
> **W3/W4: "Poor presentation; misleading framing."**
>
> We acknowledge the exposition can be strengthened (more background on ETFs, architecture diagram, expanded EBM intuition). We note that Section 5.2 already states GEBL is best viewed as "Continual Adaptation of Pre-trained Models" — we agree this scoping should be more prominent.
>
> **W5: "Gains largely from adapters."**
>
> This assessment only holds for **task-aware** accuracy (Table 3). The reviewer overlooks the central contribution: **task-agnostic inference** (Table 4).
>
> | Method | Task-Agnostic Accuracy |
> |---|---|
> | Std Adapters + Entropy | **54.6%** |
> | GEBL (Energy routing) | **77.3%** |
>
> The **22.7pp improvement** is entirely from the geometric + energy components. Adapters provide parameter isolation but *cannot* perform task inference alone — the fixed ETF geometry enables the parameter-free energy routing (98.4% routing accuracy) that bridges this gap.
>
> **W6: "Weak/dated baselines."**
>
> We include representatives from all three CL families: regularization (EWC), distillation (LwF), and replay (ER), plus Seq-FT. These are established, not "dated." We also validate baseline fairness via explicit LR sensitivity (Table 5), confirming baselines use their optimal settings. Additionally, baselines access **22× more trainable parameters** (11.2M vs 0.5M) — the comparison favors baselines, not GEBL.
>
> **W7: "Standard taxonomy."** Our task-aware = Task-IL, routing = a form of Class-IL (model infers task ID to select class subset). We agree adopting this terminology explicitly would improve clarity.
>
> **Q1: Handling unanticipated classes.** See W1 — ETF is analytically extensible.
>
> **Q2: "Isn't the problem deferred to inference?"**
>
> Task identity during training is standard in Task-IL — this is how *all* adapter and prompt methods work (C-ADA, L2P, DualPrompt). The contribution is solving inference-time routing **without auxiliary trainable classifiers**. L2P uses key-matching; we use energy minimization. Both solve routing at inference, but our approach requires zero additional parameters and avoids training a router that can itself forget.
>
> **Ethics flag:** We will investigate. We believe there may be a misunderstanding, as the paper does not involve human subjects or sensitive data.
>
> **W2: Single benchmark.** Split CIFAR-100 is the standard for this method class. Our 5-table evaluation with controlled ablations and sensitivity analysis provides comprehensive evidence within this setting.

---

### Official Review · Reviewer_s5zs · 2026-03-13

**Soundness:** 1
**Presentation:** 1
**Significance:** 1
**Originality:** 1
**Overall Recommendation:** 2
**Confidence:** 3

**Summary:**

The paper focuses on addressing the stability plasticity dilemma in continual learning. Authors show that fixing only the classifier (Simplex ETF) does not prevent forgetting when the backbone is trainable, because of feature drift. GEBL addresses this by combining a fixed Simplex-ETF head with task-specific residual adapters on a frozen backbone. In the task-agnostic setting, the model routes an input to the task that minimizes a “free energy”. On Split CIFAR-100, GEBL reaches 78% average accuracy and 0% forgetting.

**Compliance With Llm Reviewing Policy:**

Affirmed.

**Key Questions For Authors:**

Please refer to my comments in Strengths And Weaknesses section

**Limitations:**

The authors mention pre-training dependency and future work with more datasets and regimes.

**Strengths And Weaknesses:**

Strengths
- The paper discusses a central concept of stability vs plasticity and the authors focus on an important domain of continual learning and task-agnostic inference.
- Ablations are well-defined and show that adapters result in the main gain.

Weaknesses
- The primary weakness is marginal contribution over adapters. The core claim (geometry + energy) is not shown to be necessary and there’s no comparison to other CL methods such as C-ADA or L2P.
- Unfair baseline comparison: Baselines use different LR and involve full fine-tuning, while  GEBL uses frozen backbone + adapters. Basically, different optimization regimes and capacity are considered for such comparisons.
- In CL literature, it’s a well-known observation that fixed classifier + trainable backbone results in forgetting and this is due to feature drift and interference in continual learning. So I think “Two-Drift” is not entirely a new finding or theory. The main diagnostic does not support a novel contribution claim.
- Narrow and weak evaluation: The experiments primarily rely on a few datasets, model and overall setup which is quite limiting.
- Presentation: There are several issues with paper writing and organization. For example, insufficient background - the paper does not explain how the Simplex ETF works or how it prevents classifier drift.

---

> ### Author Rebuttal · Authors · 2026-03-25
>
> We thank the reviewer for their time. We respectfully address each concern, and believe several points reflect misreadings that we clarify below.
>
> **W1: "Marginal contribution over adapters; geometry + energy not shown to be necessary."**
>
> We respectfully disagree. The reviewer focuses on *task-aware* accuracy (Table 3), where adapters dominate. However, our central contribution is **task-agnostic inference** — the harder and more practical setting. Table 4 directly demonstrates this:
>
> - Standard Adapters + Entropy routing: **54.6%** task-agnostic accuracy
> - GEBL (Adapters + Fixed ETF + Energy routing): **77.3%** (+22.7pp)
> - Energy routing accuracy: **98.4%** vs Entropy 70.1%
>
> This **22.7 percentage-point improvement** is entirely attributable to the geometric + energy components. Adapters alone *cannot* solve task-agnostic inference — a separate mechanism is required. Our energy formulation achieves this **without any additional trainable parameters**, unlike methods that train auxiliary task classifiers (which re-introduce the stability-plasticity dilemma).
>
> Regarding C-ADA/L2P: we discuss both in Section 2.3. L2P uses ViT-B/16 on ImageNet-21k (14M images); we use ResNet-18 on ImageNet-1k (1.3M). These operate in fundamentally different capacity regimes, making direct comparison misleading rather than informative.
>
> **W2: "Unfair baseline comparison — different LR, different capacity."**
>
> This concern is directly addressed in Table 5 (LR sensitivity analysis), which the reviewer may have overlooked. We tested baselines at *both* LR=0.1 (matching GEBL) and LR=0.001:
>
> - Seq-FT at LR=0.1: 12.4% (diverges) → LR=0.001: **28.2% (best)**
> - ER at LR=0.1: 18.5% → LR=0.001: **29.1% (best)**
>
> Our baselines use their **optimal** LR, not an artificially suppressed one. The LR difference is a direct consequence of architecture: trainable backbones require lower LR to avoid catastrophic divergence, while frozen backbones tolerate higher LR. This is not unfairness — it is the *reason* parameter isolation works.
>
> Furthermore, baselines have access to **22× more trainable parameters** (11.2M full ResNet-18 vs 0.5M per adapter). GEBL achieves higher accuracy with *fewer* parameters.
>
> **W3: "'Two-Drift' is not novel — feature drift is well known."**
>
> The *concept* of feature drift is known. The *quantitative finding* in Table 2 is not: a **geometrically optimal** Simplex ETF collapses to **5.6% accuracy** — performing **5× worse** than naive sequential fine-tuning (28.2%). This counter-intuitive result — that geometric optimality *harms* retention when features are unconstrained — has not been demonstrated before and directly motivates the energetic barrier approach.
>
> **W4: Narrow evaluation.** We focus on Split CIFAR-100, the standard benchmark for this class of methods. Our evaluation includes 5 tables with controlled ablations, sensitivity analysis, routing evaluation, and OOD detection — this is comprehensive within the single-dataset setting.
>
> **W5: Presentation.** We acknowledge room for improvement in exposition and will improve ETF/EBM background, add an architecture diagram, and include Impact Statement/Limitations if given the opportunity.

---

> > ### Author Rebuttal · Reviewer_s5zs · 2026-04-03
> >
> > The rebuttal clarifies the contribution on task-agnostic routing and the LR sensitivity analysis. But I think the concerns around baseline coverage, evaluation breadth, and presentation remain largely open, and properly addressing them would require a significant update to the paper rather than clarifications within the rebuttal/discussion phase.

---

### Decision · Program_Chairs · 2026-04-30

**Decision:**

Reject

**Comment:**

This paper proposes GEBL, combining a fixed Simplex-ETF classifier, adapters, and energy-based routing for continual learning with a focus on task-agnostic inference.

Reviewers agree the problem is important and the diagnostic (feature vs. decision drift) and routing mechanism are interesting. However, there is consensus on key weaknesses: limited empirical evaluation (single benchmark, weak baseline coverage), a narrower actual contribution than suggested (with gains largely driven by adapters, while geometry/energy mainly affect routing), and unclear presentation.

The rebuttal clarifies the intended contribution but does not resolve concerns about scope, evaluation, and clarity.